# Promoter interactome of human embryonic stem cell-derived cardiomyocytes connects GWAS regions to cardiac gene networks

Mun-Kit Choy [1], Biola M. Javierre[2,3], Simon G. Williams[1], Stephanie L. Baross[1], Yingjuan Liu[1], Steven W. Wingett[2], Artur Akbarov[1], Chris Wallace [4,5], Paula Freire-Pritchett[2,6], Peter J. Rugg-Gunn [7], Mikhail Spivakov [2], Peter Fraser [2,8] & Bernard D. Keavney [1]

Long-range chromosomal interactions bring distal regulatory elements and promoters together to regulate gene expression in biological processes. By performing promoter capture Hi-C (PCHi-C) on human embryonic stem cell-derived cardiomyocytes (hESC-CMs), we show that such promoter interactions are a key mechanism by which enhancers contact their target genes after hESC-CM differentiation from hESCs. We also show that the promoter interactome of hESC-CMs is associated with expression quantitative trait loci (eQTLs) in cardiac left ventricular tissue; captures the dynamic process of genome reorganisation after hESC-CM differentiation; overlaps genome-wide association study (GWAS) regions associated with heart rate; and identifies new candidate genes in such regions. These findings indicate that regulatory elements in hESC-CMs identified by our approach control gene expression involved in ventricular conduction and rhythm of the heart. The study of promoter interactions in other hESC-derived cell types may be of utility in functional investigation of GWAS-associated regions.

[1] Division of Cardiovascular Sciences, The University of Manchester, Manchester M13 9PT, UK. [2] Nuclear Dynamics Programme, The Babraham Institute, Cambridge CB22 3AT, UK. [3] Josep Carreras Leukaemia Research Institute, Campus ICO-Germans Trias I Pujol, Badalona, 08916 Barcelona, Spain. [4] MRC Biostatistics Unit, University of Cambridge, Cambridge CB2 0SR, UK. [5] Department of Medicine, University of Cambridge, Cambridge CB2 0QQ, UK. [6] Division of Cell Biology, Medical Research Council Laboratory of Molecular Biology, Cambridge CB2 0QH, UK. [7] Epigenetics Programme, The Babraham Institute, Cambridge CB22 3AT, UK. [8] Department of Biological Science, Florida State University, Tallahassee, 32306 FL, USA. Correspondence and requests for materials should be addressed to M.-K.C. (email: munkit.choy@manchester.ac.uk) or to P.F. (email: pfraser@bio.fsu.edu) or to B.D.K. (email: bernard.keavney@manchester.ac.uk)

Long-range chromosomal interactions play an important role in differentiating cells, where they organise three-dimensional promoter-enhancer networks that regulate lineage-specifying developmental genes[1,2]. Distal promoter contacts are highly cell-type specific[3] and form networks of co-regulated genes correlated with their biological functions[1]. The *cis*-regulatory contact upon lineage commitment is a dynamic process that includes acquisition and loss of specific promoter interactions[4]. Cardiomyocytes differentiated from human embryonic stem cells (hESC-CMs) have become a popular model to elucidate functional genomics of cardiac disease; however, there is little information supporting the notion that genomic regions associated with adult phenotypes (such as heart rate) play important roles in hESC-CMs, which have an immature phenotype. More generally, it is essential, in the context of potential cardiac regenerative medicine approaches, to understand how promoters make specific interactions in hESC-CMs to regulate the gene expression patterns underlying the process of cardiomyocyte development. In this study we identify the promoter interactome of hESC-CMs using promoter capture Hi-C (PCHi-C)[5], which allows physical promoter-enhancer contacts in cells to be identified. We demonstrate that the promoter interactome of cardiac genes is reorganised after cardiac differentiation and cardiac enhancers are enriched in the promoter-interacting regions (PIRs). The PIRs also share the same target genes with expression quantitative trait loci (eQTLs) of cardiac left ventricles and overlap genetic variants associated with conduction and rhythm of the heart.

## Results

**PCHi-C identifies promoter interactions in hESC-CMs**. In order to elucidate the spatial promoter-enhancer organisation after the differentiation process of cardiomyocytes from hESCs, we used the PCHi-C approach[5] on three biological replicates of 20 million hESC-CMs. From the three biological replicates, we mapped and obtained 774 million valid, unique and on-target captured read pairs (63.3% capture efficiency) with Hi-C User Pipeline (HiCUP)[6], and called 179,880 significant hESC-CM

promoter–genome interactions (Supplementary Data 1) with Capture Hi-C Analysis of Genomic Organisation (CHiCAGO)[7]. Of the promoter interactions, 23.7% were common in all three biological replicates (Supplementary Fig. 1). We detected interactions between 18,159 unique promoters and 107,145 unique hESC-CM promoter-interacting regions (cPIRs) at an average degree of one promoter to 35.1 cPIRs (median = 21; maximum = 433; minimum = 1). In turn, each cPIR interacted with an average of 2.9 promoters (median = 2; maximum = 57; minimum = 1). Also, 60.5% of the interactions occurred within previously published topologically associating domains of hESCs[8]. PIRs are known to be associated with the presence of histone modifications that are hallmarks of functional activities[1,3–5,7]. We identified the locations of hESC-CM histone marks, H3K4me3, H3K27me3 and H3K36me3, by analysing published data of histone chromatin immunoprecipitation-sequencing (ChIP-seq) in hESC-CMs[9] using Model-based Analysis of ChIP-Seq (MACS)[10]. We also determined whether the genes corresponding to the promoters were transcriptionally active in hESC-CMs by analysing published RNA-seq data in hESC-CMs[11]. The active histone marks of H3K4me3 and H3K36me3 were significantly associated with transcriptionally active promoters involved in hESC-CM interactions and the repressive histone mark of H3K27me3 with the inactive ones (Fig. 1). Consistent with the reported observations that the transcriptional status of a promoter can be reflected in the histone marks of its interacting partners[4,5], H3K4me3 and H3K36me3 were also significantly enriched in the cPIRs interacting with active promoters, while H3K27me3 was enriched in those interacting with inactive ones (Fig. 1). Thus, our approach identified functionally significant promoter interactions during cardiomyocyte development.

**Cardiac VISTA enhancers are specifically enriched in cPIRs**. In order to further verify the functional roles of the cPIRs in cardiac cells, we examined whether the cPIRs intersect with enhancers that are conserved and experimentally verified in the murine heart listed in the VISTA Enhancer Browser[12]. cPIRs were found to overlap with cardiac VISTA enhancers highly significantly

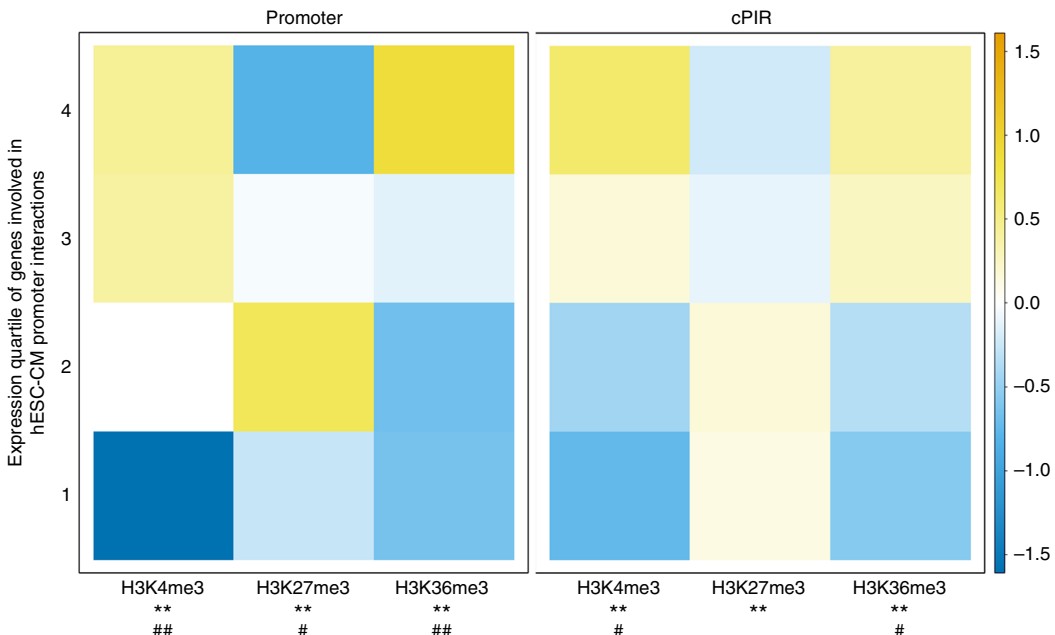

**Fig. 1** Enrichment of histone marks in gene promoters and corresponding cPIRs. Enrichment of histone marks of H3K4me3, H3K36me3 and H3K27me3 in gene promoters and corresponding cPIRs, divided into gene expression quartiles (RPKM). Histone mark enrichment is expressed as log2(proportion of enrichment in each quartile/proportion of enrichment in total). \*\*Chi-square $p < 0.01$; #Cramer's *V* > 0.1; ##Cramer's *V* > 0.2

when compared to the random background (Bonferroni-adjusted $p = 0.006$; Fig. 2a; Supplementary Data 2). The two non-cardiac control sets of enhancers, active in branchial arch and limb respectively, showed no significant overlap (Bonferroni-adjusted $p > 0.05$; Fig. 2a). By contrast, PIRs detected in hESCs significantly overlapped with VISTA enhancers of all three tested developmental tissues, reflecting the pluripotency of these cells ($p < 0.05$; Fig. 2a). The target genes of the VISTA enhancers have previously been predicted based on the enhancer-gene linear proximity alone. Using the promoter interactomic data, we mapped the VISTA enhancers to their interacting gene promoters (Supplementary Data 2). We functionally confirmed a promoter–cPIR/ VISTA enhancer interaction for VISTA enhancer mm172. This enhancer is conserved, cardiac specific in mice (Fig. 2b) and involved in heart development[13,14]. It has been predicted by VISTA to interact with the promoter of either *Ednra* or *Ttc29* based on proximity. Our promoter interactomic data showed an interaction between the human orthologue of mm172 and the promoter of *EDNRA* (Fig. 2c), a gene that is important for myocardium formation[15]. We deleted the rat orthologue of mm172 in the rat cardiac myoblast cell line H9c2 (2-1) using CRISPR-Cas9 technology and showed that the gene expression of *Ednra* was up-regulated, indicating that mm172 can function as a repressive regulatory element for *Ednra* (Fig. 2d; Supplementary Figs. 2 and 3).

**hESC-CM promoter interactome predicts target genes of eQTLs.** Since overlap with eQTLs can provide evidence for PIR regulatory functions[3], we also investigated if the promoter interactome predicts the target genes of eQTLs in two cardiac tissues, the left ventricle and atrial appendage, using publicly available data from the Genotype-Tissue Expression (GTEx) Portal[16]. Where PIRs in our data overlapped a GTEx eQTL, we ascertained, separately for cPIRs and hESC PIRs, whether the target gene of the eQTL was the same as the target gene of the PIR, and expressed the agreement as a percentage of all eQTL-overlapping PIR–gene pairs. We tested whether there was evidence for enhanced agreement within the cPIRs or hESC PIRs compared with the random background. Previous data indicate that higher Hi-C read counts provide more robust support for interactions between distal regulatory elements and target genes[17], and therefore we explored the effect of different read count cut-offs on the agreement between eQTL–gene and PIR–gene pairs. As expected, the agreement percentage increased with sequencing read counts of promoter interactions (Fig. 3). The highest significance was achieved with the top 20% hESC-CM promoter interactions and left ventricular eQTLs (Bonferroni-adjusted $p < 0.05$), with a similar trend observed in the atrial appendage (Fig. 3). The higher significance of agreement between target genes of cPIRs and eQTLs in the left ventricle is consistent with the finding that hESC-CMs typically mature into ventricular-like cardiomyocytes[18].

Of the 1264 top 20% PIR–gene interactions overlapping with lead left ventricular eQTLs (in strong linkage disequilibrium blocks (LD; $r^2 > 0.8$)), 16.5% of them target the same genes as the eQTLs. eQTLs are linked to trait-associated single-nucleotide polymorphisms (SNPs)[19] and therefore are important information for functional elucidation of genetic findings. Given that the top 20% cPIRs for read counts showed the most significant agreement of target genes with ventricular eQTLs, we reasoned that these top 20% of promoter interactions represented the most relevant subset mediating eQTL–gene interactions.

**Cardiac promoter networks reorganise upon differentiation.** Within the 35,960 top 20% hESC-CM promoter interactions,

6376 unique promoters interacted with 27,123 unique cPIRs at an average degree of one promoter to 13.7 cPIRs (median = 10; maximum = 74; minimum = 1). However most cPIRs interacted with only one promoter (average = 1.8; median = 1; maximum = 11; minimum = 1). Also, 89.9% of the interactions occurred within previously published topologically associating domains of hESCs[8]. The target genes of the top 20% hESC-CM promoter interactions included key regulators of cardiovascular development and function (Supplementary Table 1). The top 20% hESC-CM promoter interactions were found to constitute 3547 sub-networks with connected nodes using Cytoscape[20]. The subnetwork with the highest average degree (5.85) had 40 nodes, including 12 promoter nodes and 28 cPIR nodes, and 117 edges (Fig. 4a; Supplementary Data 3). The 12 promoter nodes represent 17 genes in the proximity of *MYH6* (most connected) and *MYH7* genes (Fig. 4a) and the *MYH6-MIR208A-MYH7-MIR208B* network is known as myomiR network that regulates myosin heavy chains in the heart during hypertrophic responses[21]. Given the paucity of promoter interactions in *MYH6* region observed in hESCs (~5 times less; Fig. 4b), the highly connected cluster of interactions in this locus observed in hESC-CMs has likely emerged through dynamic reorganisation[4] of chromosomal interactions upon differentiation. *MYH6* and *MYH7* are key genes in cardiac development and function. Mutations in *MYH6* cause congenital heart diseases (CHDs), in particular defects in the atrial and ventricular septum, via dysfunction of cardiac myofibrils[22]. *MYH7* mutations are a major cause of hypertrophic cardiomyopathy, and have also been reported in CHD families[22]. The identification of promoter interaction networks in hESC-CMs involving key cardiac genes, such as *MYH6* and *MYH7* in our data, illustrates the potential of this approach to map the relationships between genes and regulatory elements in genomic regions of cardiac importance.

**hESC-CM promoter interactions are associated with heart rate.** Promoter interactions detected in trait-relevant tissues hold the clues to the putative target genes of non-coding genome-wide association study (GWAS) SNPs[23]. We therefore explored if the hESC-CM promoter interactome was associated with any of the three cardiovascular phenotype groups by considering their GWAS profiles: (1) CHD[24,25]; (2) coronary artery disease (CAD; http://www.cardiogramplusc4d.org)[26]; and (3) cardiac conduction and rhythm disorders (CRD)[27]. Our prior hypothesis was that we would find an association between the hESC-CM promoter interactome and CRD, which is a phenotypic group largely deriving from dysfunction that involves myocardial cells, but not for CHD or CAD. For each GWAS, we obtained the $p$ values of SNPs located in cPIRs or hESC PIRs and constructed QQ plots against averaged $p$ values of SNPs located in 1000 sets of randomised cPIRs or hESC PIRs (Fig. 5a). For SNPs located in cPIRs (cPIR SNPs), the highest inflation was observed in GWAS $p$ values of CRD (inflation factor, $\lambda = 1.237$). On the other hand, SNPs located in hESC PIRs showed the highest inflation in GWAS $p$ values of CAD (inflation factor, $\lambda = 1.250$). We then investigated whether all the inflated SNPs ($p < 0.001$) from each of these GWAS studies overlapped significantly with the top 20% cPIRs or hESC PIRs. Using GoShifter (https://github.com/ immunogenomics/goshifter)[28], we observed a significant overlap for only CRD GWAS SNPs (Bonferroni-adjusted $p = 0.045$). This indicated that cPIRs are enriched for regulatory regions controlling genes involved in cardiac rhythm, a phenotype in which cardiomyocytes directly participate.

Some of the significant cPIR SNPs (at the suggestive threshold of $p < 1 \times 10^{-5}$) associated with CRD were interacting with promoters of known heart rate-associated genes[27] such as

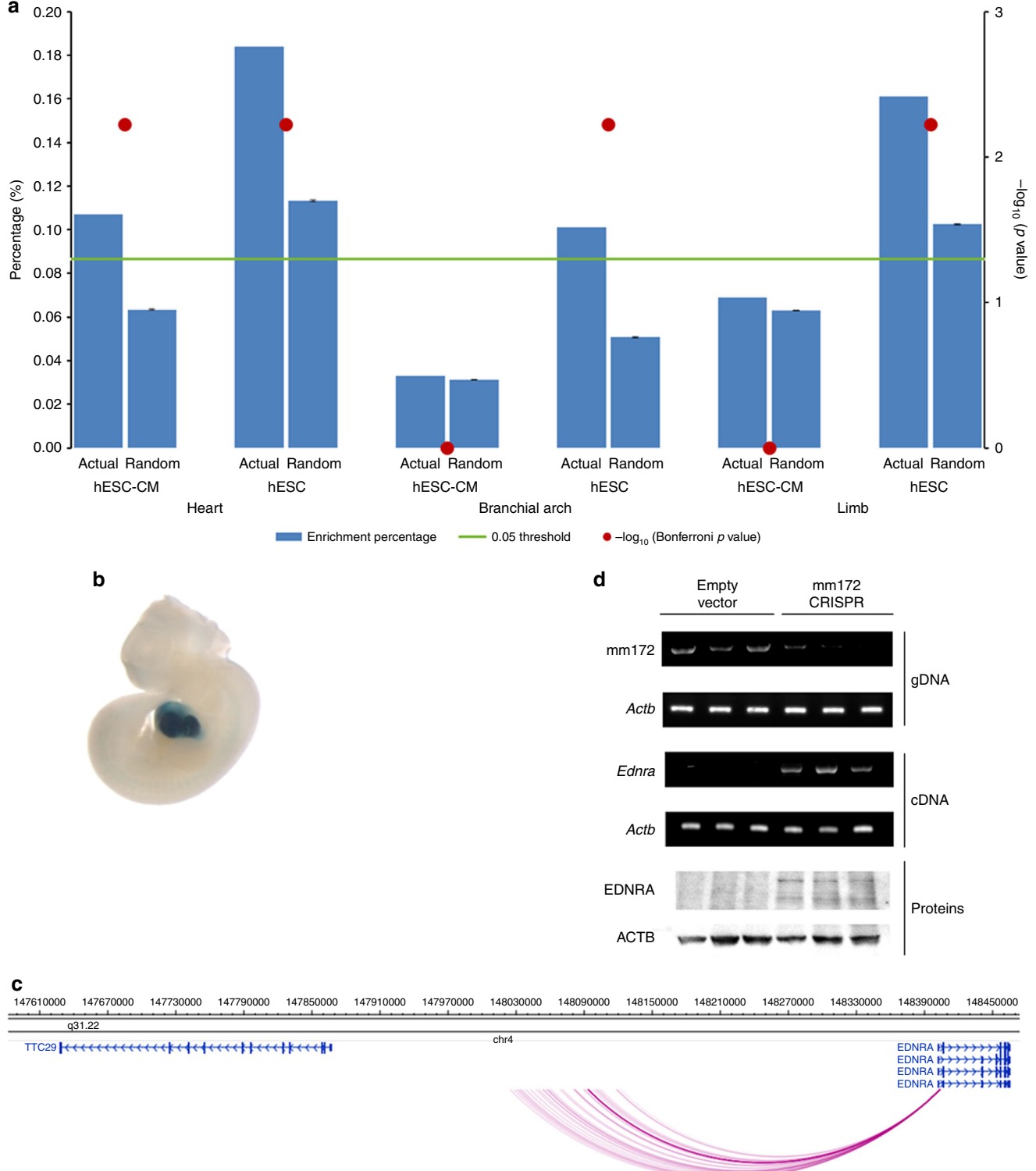

**Fig. 2** Enrichment of cardiac VISTA enhancers in cPIRs. **a** Percentages of cPIRs or hESC PIRs overlapping with VISTA enhancers of the heart, branchial arch and limb. Results obtained with randomised PIRs (1000 permutations) are shown as controls. The p values were generated from permutation tests performed to compare observed vs. randomised PIRs and corrected for multiple testing using Bonferroni procedure. **b** Image of mm172-reporter staining of an E11.5 mouse embryo obtained from the VISTA enhancer browser (https://enhancer.lbl.gov/)[14]. **c** Promoter–cPIR interactions of hESC-CM occurring between the promoters of *TTC29* and *EDNRA* (thick arc line represents the interaction between *EDNRA*'s promoter and mm172). **d** PCR using gDNA/cDNA to detect mm172 deletion/*Ednra* expression and immunoblotting to detect EDNRA (glycosylated) were performed to compare rat cardiomyoblast H9c2 cell lines transfected with the empty plentiCRISPRv2 plasmid or the same plasmid cloned with two gRNA sequences to target mm172. *Actb*/ACTB is the loading control

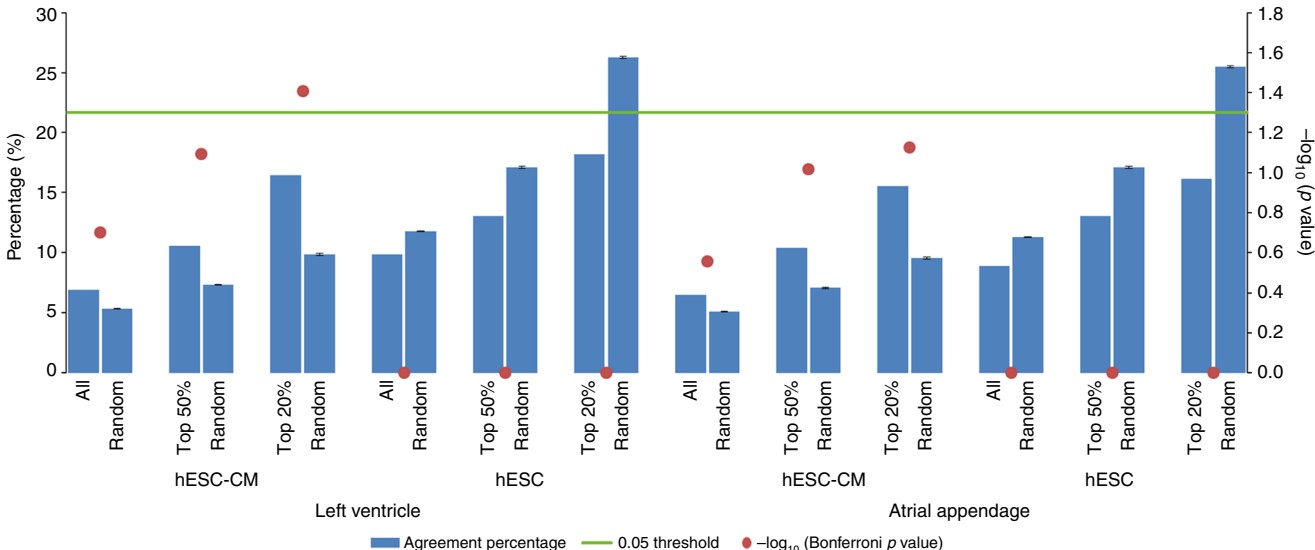

**Fig. 3** Agreement percentages of eQTL–gene and PIR–gene pairs. Agreement percentages of eQTL–gene (GTEx) and PIR–gene pairs (either cPIRs or hESC PIRs), filtered to the top 20% and 50% of read counts, in the left ventricle and atrial appendage. Results obtained with randomised PIRs (1000 permutations) are shown as controls. The p values were generated from permutation tests performed to compare observed vs. randomised PIRs and corrected for multiple testing using Bonferroni procedure

*CCDC141* on chromosome 2, *GJA1* on chromosome 6 and *MYH6* on chromosome 14 (Fig. 5b; Supplementary Data 4). With the promoter interactomic data, the interactions of variants within a GWAS signal can be further identified, for example, the complex *MYH6* interaction network in *MYH6* region and *CCDC141-SESTD1* interactions in *CCDC141* region. Interacting partners of SESTD1 such as TRPC (transient receptor potential-canonical) channels are important regulators of the calcium-dependent hypertrophy pathway[29,30]. The significant cPIR SNPs also coincided with published regions of *TFPI* on chromosome 2, *SLC35F1* on chromosome 6 and *HCN4* on chromosome 15 but, upon integration of the chromosomal interaction information, those regions were found interacting with *ZSWIM2*, *PLN* and *NPTN* respectively (Fig. 5b; Supplementary Data 4). *PLN* is a highly credible candidate gene because PLN is a crucial regulator of cardiac contractility[31]. Neuroplastin (NPTN) forms a complex with plasma membrane $Ca^{2+}$ ATPases to modulate calcium homoeostasis[32,33]. The Np55 variant of NPTN is expressed in the heart[34] and the disruption of NPTN has been shown to cause increased heart weight in the International Mouse Phenotyping Consortium database (IMPC; https://www.mousephenotype.org/) [35]. ZSWIM2 is an E3 ubiquitin-protein ligase, a family of proteins that may play a role in degrading mal-processed ion channel proteins in the heart[36]. It has been reported that genes some distance from a strong candidate locus harbouring a GWAS "hit" SNP may have an important influence on complex phenotypes[37]. For example, polymorphism in the *FTO* gene are the strongest GWAS signals for obesity, but the functional mechanism of the association appears chiefly due to the effects of the SNPs on two neighbouring genes, *IRX3* and *RPGRIP1L*[37]. Therefore, promoter interactions of the neighbouring genes within a GWAS signal may provide crucial information to pinpoint the true target genes.

In addition, we identified a peak in chromosome 19 with significant cPIR SNPs interacting with *ACTN4–CAPN12* (Figs. 5b and 6a; Supplementary Data 4). This peak did not reach conventional significance ($p < 5 \times 10^{-8}$) in the GWAS study[27]. However, by incorporating eQTL data from the GTEx Portal[16], the significant cPIR SNPs in chromosome 19 in our data were shown to be significant eQTLs for *ACTN4* and *CAPN12* in the left ventricle of human hearts (Fig. 6b). Functionally, calpains may

play a role in atrial fibrillation[38] and ACTN4 may promote muscular differentiation as a transcriptional regulator[39]. One candidate explanation for "missing heritability" in genetic studies of complex traits is that many biologically significant (and potentially druggable) genetic effects are of insufficient size to be detected at conventional significant levels even in large GWAS studies[40]. By focusing on interactions in regions of borderline significant genetic associations, we were able to identify potentially physiologically relevant target genes of GWAS SNPs through functional rather than proximity considerations. This may, in the future, facilitate the identification of new genes influencing cardiac conduction and rhythm disorders.

## Discussion

The hESC-CM interactomic map of gene promoters generated in this study not only enabled us to understand the promoter–genome networks of established cardiac genes but also those of a number of novel genes possibly playing a role in cardiac development and function. We mapped known conserved cardiac-specific VISTA enhancers to their target genes using the promoter interactome. We validated one of the VISTA enhancers involved in cardiac development, mm172, by deleting it using CRISPR-Cas9 and showed that it indeed was a regulatory element of its interacting gene *EDNRA*, which is a critical gene in chamber myocardium formation during heart development[15]. Somewhat unexpectedly, we observed that deleting this enhancer resulted in an increase of *EDNRA* expression, consistent with a number of repressive lineage-specifying promoter interactions detected in the pluripotent state[2] and in early hESC-derived neural progenitors[4]. There remains a possibility that silencers of the same gene could be disrupted as well when a cPIR/enhancer was targeted by CRISPR-Cas9 since enhancers and silencers can be located in the same locus or clustered in the same locus control region[41]. Further work will be required to elucidate this possibility.

The GWAS SNPs associated with cardiac conduction and rhythm disorders were enriched in cPIRs. While the target genes of GWAS SNPs are usually identified by gene proximity, it is known that non-coding variations can influence genes situated at

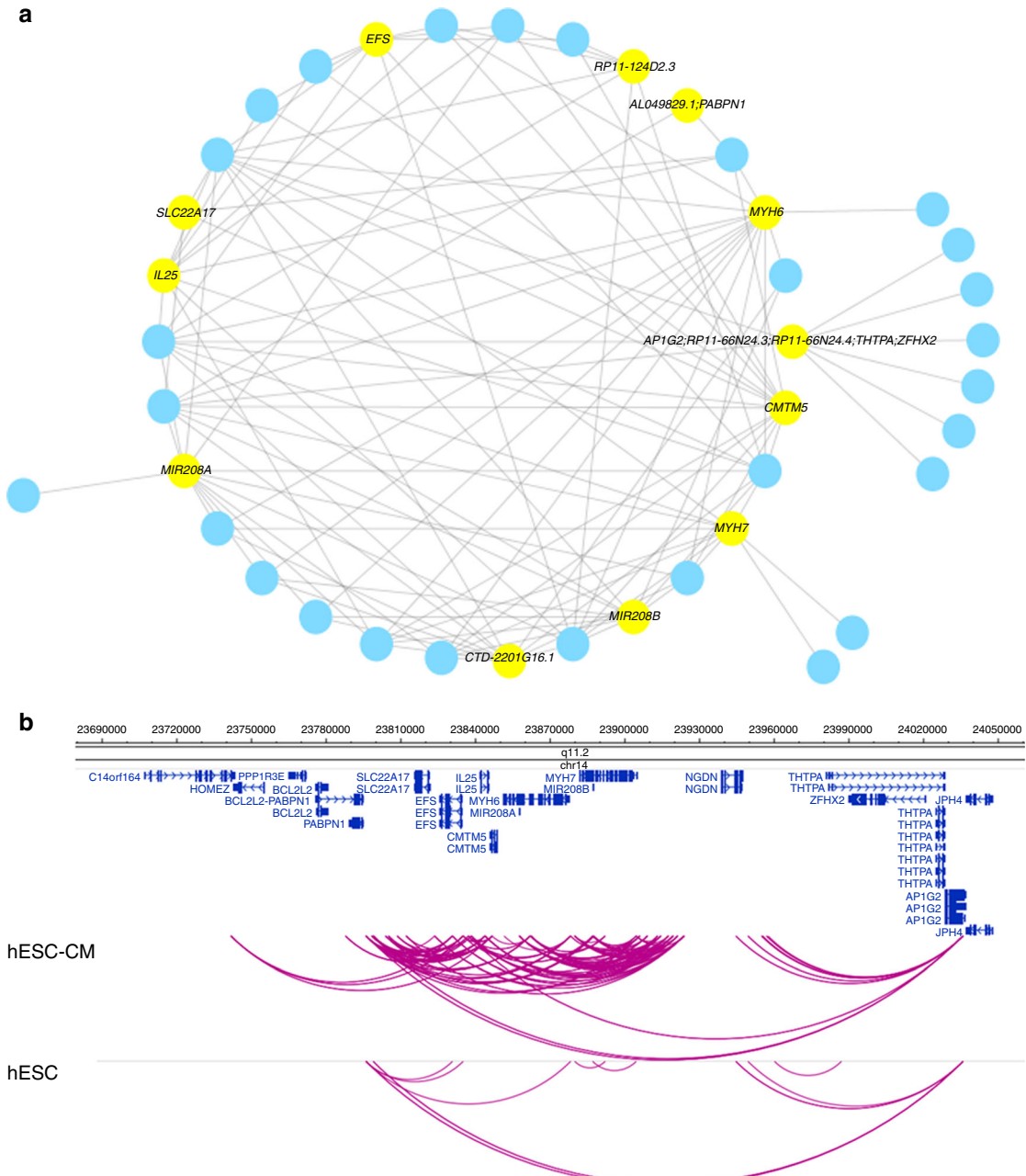

**Fig. 4** Network analyses of the top 20% hESC-CM promoter interactions after differentiation. **a** The promoter–cPIR subnetwork with the highest average degree among the top 20% promoter interactions for read counts, occurring in the *MYH6* region having 40 nodes, including 12 promoter nodes (yellow) and 28 cPIR nodes (blue), and 117 edges. **b** Reorganisation of the *MYH6* network between the states of hESCs and hESC-CMs

substantial distance, rendering selection of a candidate in a region problematic. Using promoter interactions in hESC-CMs, we were able to identify distal target genes of GWAS SNPs for a phenotype relevant to the specific cell type and new genomic regions associated with cardiac conduction and rhythm disorders. The hESC-CM promoter interactome, rather than the hESC promoter interactome, also had a significantly higher likelihood, compared with the random background, mapping to the target genes of the left ventricular eQTLs located within the cPIRs. Taken together with the GWAS findings, it suggests that hESC-CMs are a relevant model to study ventricular conduction and rhythm. Recently, an "omnigenic" model of complex disease involving few "core" genes and many "non-core" genes influencing a particular phenotype has been proposed[42]; the most complex interactomic subnetwork of cardiomyocytes in our data were represented

prominently among GWAS "hits" for the heart rate, which would be in keeping with this model. Moreover, our integration of interactomics with GWAS data suggest that similar approaches to that we present may prove particularly useful to identify "core" genes. This approach may be extended to other hESC-derived cell types to functionally elucidate GWAS regions.

## Methods

**The hESC-CM differentiation**. The hESC line WA09 (H9) was obtained from WiCell, maintained and cultured in monolayers according to WiCell Feeder-Independent Pluripotent Stem Cell Protocols using mTeSR1 Medium (SOP-SH-002). The hESC-CM differentiation was performed on WA09 hESC monolayers (passage 32) at 80–100% confluence using PSC Cardiomyocyte Differentiation Kit (Gibco; A2921201). After the treatments of Cardiomyocyte Differentiation Medium A and B (2 days each), the cells were allowed to recover in Cardiomyocyte Maintenance Medium for a day. The cells were then passaged (1:1.5 ratio) into

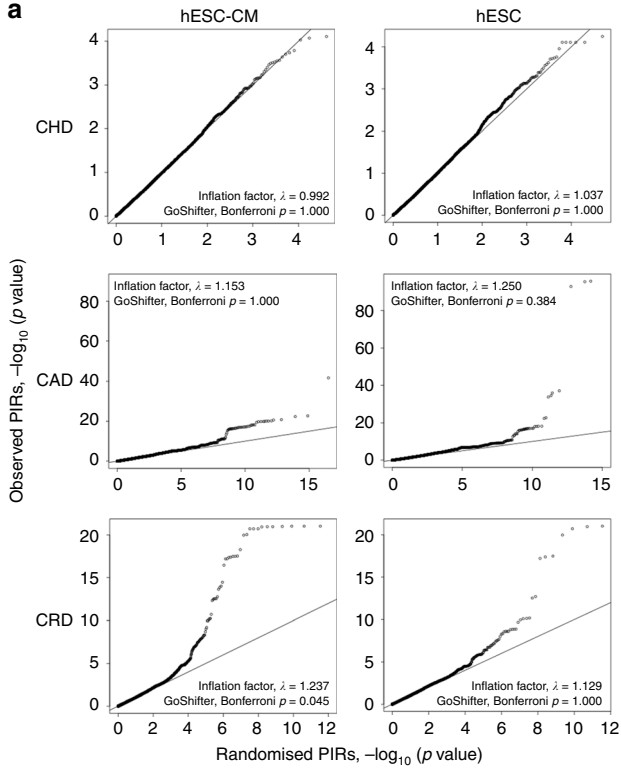

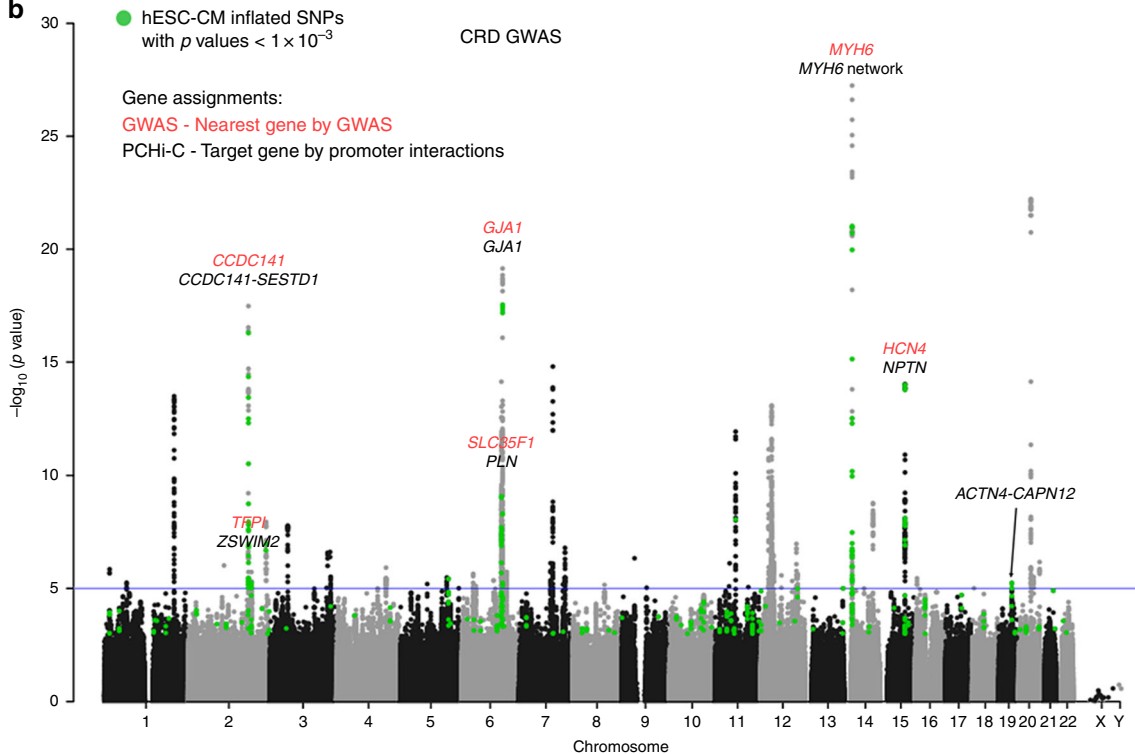

**Fig. 5** Association of hESC-CM promoter interactions with cardiac conduction and rhythm disorders. **a** QQ plots of –log$_{10}$($p$ values) corresponding to SNPs located in the top 20% cPIR or hESC PIRs of three GWAS studies, congenital heart disease[25, 26] (CHD; total SNPs = 500K; hESC-CM SNPs = 19K; hESC SNPs = 19K), coronary artery disease[26] (CAD; total SNPs = 9.5 M; hESC-CM SNPs = 289K; hESC SNPs = 328K; http://www.cardiogramplusc4d.org) and cardiac conduction and rhythm disorders[27] (CRD; total SNPs = 2.5 M; hESC-CM SNPs = 85K; hESC SNPs = 80K), against randomised PIRs (average of 1000 permutations). Significance of overlaps between GWAS SNPs ($p < 0.001$) and the top 20% cPIRs or hESC PIRs were tested using GoShifter[28]. **b** Manhattan plot of CRD GWAS[27]. Green spots indicate inflated SNPs in the cPIR/randomised PIR QQ plot with $p < 0.001$. Red gene names are nearest genes to the SNPs in the significant peaks assigned by the GWAS and black gene names are target genes interacting with the cPIRs containing the significant SNPs ($p < 1 \times 10^{-5}$)

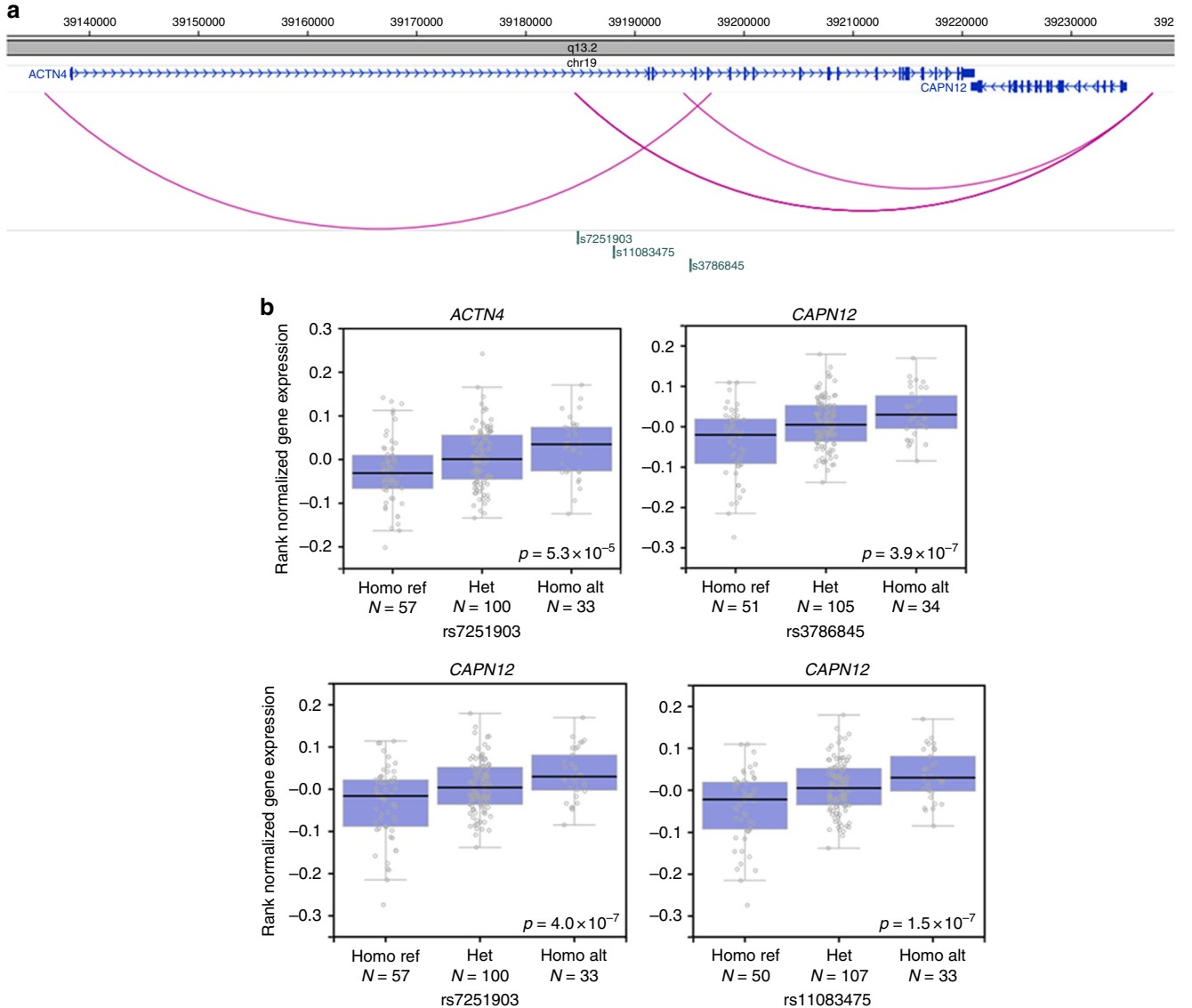

**Fig. 6** Promoter interactions and eQTLs of *ACTN4–CAPN12* peak in CRD GWAS. **a** Genomic positions and promoter interactions of cPIR SNPs in *ACTN4–CAPN12* peak of CRD GWAS. **b** Left ventricular eQTL box plots (obtained from GTEx Portal V6p; https://www.gtexportal.org/) of cPIR SNPs and target genes in *ACTN4–CAPN12* peak of CRD GWAS. Gene expression levels were stratified by the genotypes and the reference allele was associated with reduced expression levels. "Homo Ref" refers to the homozygous reference allele genotype; "Het" refers to the heterozygous genotype; and "Homo Alt" refers to the homozygous alternate allele genotype

plates coated with fibronectin (12.5 μg/ml in 0.02% gelatin; Sigma) and allowed to grow for 6 days in RPMI1640-B27 minus insulin (Gibco) before changing to RPMI1640-B27 complete (Gibco) for further 2 days. The cardiomyocytes were then enriched in glucose-depleted-lactate-enriched medium (Dulbecco's modified Eagle's medium (DMEM) without glucose (Gibco; 11966025); 4 mM lactate (Sigma); non-essential amino acids (Gibco); GlutaMAX (Gibco)) for 6 days and recovered in Iscove's modified Dulbecco's medium (Gibco) containing 20% foetal bovine serum (FBS) for 2 days. The hESC-CMs were cultured in RPMI1640-B27 complete (Gibco) for 7 more days before the quality check (at least 80–90% purity by flow cytometry and quantitative polymerase chain reaction (qPCR) using myosin heavy chain and TNNT2 as markers) and the fixation for PCHi-C.

**PCHi-C**. PCHi-C was performed as previously described[3] and summarised as follows: hESC-CMs were fixed by 2% formaldehyde (Agar Scientific; R1026) for 10 min at room temperature, quenched by cold 0.125 M glycine (5 min at room temperature followed by 15 min on ice), flash frozen in liquid nitrogen and stored at − 80 °C. Hi-C libraries were prepared with in-nucleus ligation and PCHi-C was performed with Agilent SureSelect Target Enrichment System. Then, 22,076 *Hin*dIII fragments containing a total of 31,253 annotated promoters for 18,202 protein-coding and 10,929 non-protein genes according to Ensembl v.75 (http://grch37.ensembl.org) were captured. A post-capture PCR amplification step was carried out

using PE PCR 1.0 and 2.0 primers with four PCR amplification cycles. High-throughput sequencing reactions were performed on the Illumina HiSeq2500 platform. Sequencing reads were processed and mapped with HiCUP[6] and PCHi-C interaction was called using CHiCAGO[7] with default parameters.

**Data processing and statistical analyses**. After calling the significant PCHi-C interactions with CHiCAGO[7], the biologically replicated read counts of the promoter interactions in hESC-CMs (three biological replicates) and those previously obtained from WA09 hESCs (two biological replicates; Gene Expression Omnibus (GEO); GSE86821; in-solution ligation)[4] were normalised using DESeq2 (https://bioconductor.org/packages/release/bioc/html/DESeq2.html; local dispersion fit)[43] and averaged. The peak calling for published H3K4me3, H3K27me3 and H3K36me3 histone ChIP-seq (GEO; GSE35583) of hESC(WA07[H7])-CM (14 days; one biological replicate each)[9] was performed using MACS 2.1.0 (peak mode for H3K4me3, and block mode for H3K27me3 and H3K36me3; q < 0.05; mfold = 10, 30; band width = 300)[10]. The RPKM (reads per kilobase per million mapped reads) for each gene was calculated from published RNA-seq data (GEO; GSE69618) of hESC(WA01[H1])-CMs (10 days; four biological replicates)[11]. For significant tests of enrichment/agreement percentages, randomly generated promoter–PIR interactions were created to retain the distribution of interaction distances of the observed promoter–PIRs in each set. For each promoter bait, the

relative positions of its PIRs were calculated and randomly transplanted to another randomly selected promoter bait. For each observed set, 1000 random sets were generated with each set containing the same number of interactions as the observed set, and a one-tailed permutation test was performed. Each random PIR set was then assessed for its enrichment percentage of VISTA enhancers or agreement percentage of eQTLs in the same way as the observed data and used for comparisons. For the eQTL analysis, single-tissue cis-eQTL data were downloaded from GTex (V6p; https://gtexportal.org/home/)[16]. For each tissue, lead SNP-gene pairs were selected based on them having the lowest p values within each LD block calculated using the Ensembl API (1000 genomes phase 3; EUR population; $r^2 >$ 0.8; window size = 500 kb)[44]. To assess the agreement percentage of eQTL and PIR target genes for each set of hESC-CM or hESC promoter interactions, the gene promoters that each PIR interacts with were compared to the target genes of eQTLs that were found to be located within the PIR region. The agreement percentage is then expressed as the percentage of PIRs containing at least one eQTL targeting the same gene as the PIR among all eQTL-overlapping PIR–gene pairs. GTex eQTLs were limited to those SNP-gene pairs occurring within one megabase distance and therefore promoter–PIR interactions occurring more than one megabase distance were filtered out for the eQTL agreement assessment[45]. For GWAS/QQ plot analyses, the overlap of PIRs with GWAS SNPs was determined for each set of observed hESC-CM and hESC promoter interactions. The same was done for each of the 1000 randomly generated interaction sets for each observed set. The p values of the observed overlapping SNPs and those overlapping the randomly generated PIRs were compared, first by interpolating the lists of p values to the shorted list (observed or random) and then taking the mean of each rank-ordered p value list across the 1000 permutations based on QQperm package (https://cran.r-project.org/web/packages/QQperm/index.html). The inflation factor (lambda, λ) of observed/random QQ plots were estimated using QQperm (estlambda2) as well. All sequenced reads were mapped to the human reference genome (build hg19/GRCh37).

**CRISPR-Cas9 and lentiviral transduction**. The guide RNA (gRNA) sequences (gRNA1-GGA CCA TAA CTC AAG CAG GGC AGG; gRNA2-GGC TGT GAC TTT GAT GGG CCA AGG) used to delete VISTA enhancer mm172 in rat cardiomyoblast H9c2 2-1 (Sigma; cultured in DMEM (Gibco) supplemented with 10% FBS and 1× penicillin–streptomycin at 37 °C and 5% CO₂) were cloned into plentiCRISPRv2 (Addgene; plasmid 52961) according to published paired gRNA design[46] using human U6 promoter for gRNA1 and murine one for gRNA2. The cloned plentiCRISPRv2 and second-generation packaging plasmids psPAX2 (Addgene; plasmid 12260) and pMD2.G (Addgene; plasmid 12259) were co-transfected into HEK293T cells (cultured in DMEM (Gibco) supplemented with 10% FBS and 1× penicillin–streptomycin at 37 °C and 5% CO₂) in the proportion of 4:3:2 using Lipofectamine 2000 (Invitrogen) for lentivirus production. Lentivirus-containing media were collected 48 h after transfection, centrifuged to remove cell debris and 0.45 μm filtered prior to infecting H9c2 cell monolayers in the presence of 10 μg/ml polybrene (Millipore). At 24 h post infection, the transduced H9c2 cells were selected in 1 μg/ml puromycin media for 48 h.

**Polymerase chain reaction**. Genomic DNA (gDNA) was extracted from cells using PureLink Genomic DNA Mini Kit (Invitrogen) and total RNA using RNAqueous-Micro Kit (Ambion). First-strand complementary DNA (cDNA) was synthesised from DNase (Promega) treated total RNA using M-MLV Reverse Transcriptase (Promega) and both random hexamer and oligo(dT) primers (Promega). PCR was performed using OneTaq 2×Master Mix (New England BioLabs) and the following primers: (1) mm172 (forward TGATTGGTAGAGGAGAGCGG; reverse CAAGGCTGCATCACTCTGAC; product length 2766 bp (gDNA)); (2) Ednra (forward AGCCTCTCTCTGATCCAACG; reverse CCATA-GAACTGCACGGAAGC; product length 59804 bp ((gDNA) or 1892 bp (cDNA)); (3) Actb (forward CTATGTTGCCCTAGACTTCG; reverse AGGTCTTTACG-GATGTCAAC; product length 318 bp (gDNA) or 228 bp (cDNA)); and (4) plentiCRISPRv2 (forward CGCAAATGGGCGGTAGGCGTG; reverse AAT-CATGGGAAATAGGCCCTC; product length 1857 bp (gDNA/plasmid DNA)).

**Immunoblot analysis**. Whole-cell lysates were prepared by resuspending cells in RIPA buffer containing 5× cOmplete Mini Protease Inhibitor Cocktail (Sigma) and homogenised with a 19-gauge syringe. Equal amounts of protein samples were boiled in Laemmli buffer, resolved by a Mini-PROTEAN TGX Precast Gel (Bio-Rad), transferred to a nitrocellulose membrane by a Trans-Blot Turbo Transfer System (Bio-Rad), incubated with primary antibodies against EDNRA (Abcam; ab117521; 1:2500 dilution; 48 kDa and glycosylated EDNRA 60–90 kDa[47]) and ACTB (Cell Signalling; #8457; 1:1000 dilution; 45 kDa), and subsequently with a goat anti-rabbit horseradish peroxidase secondary antibody (Pierce; 1:1000 dilution). The membrane was incubated with SuperSignal West Femto Maximum Sensitivity Substrate (Thermo Scientific) before being imaged with a Molecular Imager ChemiDo XRS+System (Bio-Rad).

**Data availability**. The PCHi-C data for hESC-CMs have been deposited in Gene Expression Omnibus (GEO) with accession number GSE100720 and Capture HiC Plotter (CHiCP; https://www.chicp.org/).

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

## Acknowledgements

We thank Heather J. Cordell and Ruth J. Loos for sharing their GWAS data. We thank David A. Eisner, Andrew W. Trafford, Luigi A. Venetucci and Yatong Li for verifying the electrophysiological phenotypes of our hESC-CMs. We also thank members of The University of Manchester, especially David Talavera and Sabu Abraham, and The Babraham Institute who provided constructive comments and shared materials/methods. This work was supported by the following grants: BHF (CH/13/2/30154), BHF (RG/15/12/31616), BBSRC (BB/J004480/1), MRC (MC_UP_1302/5, MR/L007150/1) and Wellcome Trust (097820/Z/11/A, WT107881). M.-K.C. is a Life Member of Clare Hall, Cambridge, S.L.B. was supported by FS/16/58/32734 BHF 4-Year PhD Studentship Programme, Y.L. was supported by The University of Manchester PUHSC Alliance and China Scholarships Council and A.A. was supported by HEFCE. The Genotype-Tissue Expression (GTEx) Project was supported by the Common Fund of the Office of the Director of the National Institutes of Health, and by NCI, NHGRI, NHLBI, NIDA, NIMH and NINDS.

## Author contributions

M.-K.C., B.M.J., M.S. and P.F. conceived the idea and designed the study. M.S., B.D.K. and P.F. secured funding for experiments and sequencing. M.-K.C., B.M.J., S.L.B., Y.L., P.J. R-G., P.F. and B.D.K. performed or supervised laboratory work. M.-K.C., S.G. W., S.W.W., A.A., C.W., P.F-P., M.S. and B.D.K. performed or supervised statistical analyses. M.-K.C. wrote the first draft. All authors commented critically on and revised the draft.

## Additional information

**Competing interests:** The authors declare no competing interests.

