## [Peer Review File · Nature Communications]

Reviewer #1 (Remarks to the Author):

Dr Choy & colleagues perform promotor capture Hi-C in human ESC-derived cardiomyocytes, and annotate just over 5k promotor-genome interactions.

Since these interactions are highly cell-type specific, this provides a valuable new annotation of the (human) non-coding genome of relevance and value particularly to the cardiovascular community.

The authors demonstrate the value of this cell-specific atlas by annotating relevant GWAS hits and eQTLs, and identifying physical interactions between tagging SNPs and promoters of genes that identify new candidate genes for (some of) the traits analysed.

Comments

My only major concern is that the primary comparison is between newly generated hESC-CM data and historical hESC data, here described by citation only. It looks like these were derived from the same ESC lineage (WA-09), and there is substantial authorship overlap so I presume much of the experimental workflow is directly comparable, but it would be helpful to reproduce a little of the original methodological detail here for the readers convenience and reassurance (e.g. so that it is apparent that these are the same source lineages without referring back to ref 5).

Please also clarify what QC was done to control for any batch effects between libraries prepared at different times (and different place?) or other technical differences that might affect the primary analyses.

Minor comments

These are mainly comments on data presentation for the authors' consideration

Table S1 - what is the background gene list used for pathway analysis? all genes expressed in hESC-CM?

Figure 1a — the legend could be more descriptive. In first panel (promotor) are distances either side of promotor (1kb each side), a span centred on the promotor, or upstream only?

Presumably downstream is unidirectional downstream of UTR.

What is the source of the genome annotations used by CEAS? I note that this gives 46.2% of the genome as “genic” (including introns), and 42.4% as intronic, which I think is at the upper limit of the range of estimates, and will depend on the gene models used for annotations. Suggest give CEAS version and source of annotations.

Figure 1b - Personally I do not find the heat map a very helpful representation of 8 numbers here - difficult to compare differences by eye, and plot is not grayscale-safe if printed. Not sure if color palette is red/green color-blind safe?

Figure 2a - top panel: consider adding a track with a color-coded marked that discriminates the promoters from PIRs.

lower panel - could similarly color code PIR vs promoter for clarity. Are the arrows meant to signify directionality (would expect them to be towards the gene promoter if an enhancer/repressor?)?

Have you considered providing a URL to a hosted track to make this browsable on UCSC or ensembl? (also fig 2b)

line 81 - “a number of cardiac VISTA enhancers were found to overlap” - what proportion of your PIR overlap with VISTA enhancers, and are these statistically enriched? A brief description of the method used to define VISTA enhancers, to contrast with the methods used here, might be useful for some readers.

I would add a comment acknowledging that the only region selected for validation in the wet lab was also annotated by VISTA - a broader set of wet lab validations would give better sense of whether there is an important false positive rate (though enrichment analyses provide evidence for biological relevance)

line 92 - “edna” should be “ednra”

line 101 - though whole blood eQTLs are only “marginally significant”, they are still enriched - and presumably this was anticipated as a negative control. Suggest expanding the comment here (e.g. after bonferroni correction it more enriched than atrial tissue that would be expected to be more similar.

Methods - much of the methodology is “as previously described” (refs 3 and 5). I think this could bear some repetition (in condensed form) here for convenience, but leave to your discretion.

As above - confirm the origins of the hESC data used for comparison, and methods to control for batch effects.

Data & code availability - The accession GSE100720 is private and scheduled for release on 3rd July 2020. I did not receive a reviewer’s secure token to preview this material, so have not reviewed this content.

Both the raw and processed data should be available for reproducibility: I am not sure whether both have been deposited.

The code should also be available along with the data - I am not sure whether GEO supports this - if not then would suggest using a GitHub “release” or equivalent. This can be released “as is” in its working form without any special presentation for publication

Reviewer #2 (Remarks to the Author):

This is a concise report presenting new promoter capture Hi-C data (PChi-C) in cardiomyocytes derived from human embryonic stem cells (hESC-CM). The authors show that many of the promoter-interacting DNA segments (PIRs) specific to hESC-CM (cPIRs) act as distal regulatory elements of the target genes inferred from the promoter interactions. Furthermore, the authors used the PIRs to focus on SNPs associated with cardiac phenotypes via GWAS, providing a better prediction of likely target genes and enabling discovery of significant association of SNPs at a locus not previously associated with cardiac conductance and rhythm disorders. This use of promoter interaction data to filter potentially interesting SNPs reduces the multiple hypothesis testing burden and allows discovery of associations that fall below genome-wide significance. The authors make a strong point that the approach followed in this paper should be broadly applicable.

The following are points for improving the manuscript.

(1) p. 1, lines 18-19: Characterized enhancers are genomic regulators, but eQTLs and tag SNPs associated with traits by GWAS are genetic variants potentially associated with regulatory elements.

(2) The authors should point out that candidates for distal gene regulatory elements (ascertained by epigenetic features such as chromatin accessibility and histone modifications) are enriched for phenotype-associated SNPs discovered via GWAS, with appropriate references.

(3) Fig. 1b: These results would be displayed more clearly as a graph of enrichment and depletion rather than a heat map. The legend says that the heat maps shows the enrichment of hESC-CM specific H3K4me3 enrichment in promoter, but it is not clear what is being plotted for the cPIRs.

(4) Fig. 2a and 2b: In the genome browser views, the positions of promoter vs. cPIR nodes are not clear. Showing the "promoter" DNA fragment "baits" would help, but it would be best to also include distinctive icons for promoter vs cPIR.

(5) p. 3, line 90: Presumably the authors deleted the rat ortholog of mm172.

(6) p. 5, lines 159-161: The authors state that the hESC-CM specific promoter interactome has a higher success rate in mapping target genes of particular eQTLs, but they should state the method or resource to which they are comparing.

Signed,

Ross Hardison, Penn State University

Reviewer #3 (Remarks to the Author):

The authors performed promoter capture Hi-C (PCHiC) in cardiomyocytes differentiated from human embryonic stem cells (hESC-CMs) to identify distal promoter contacts, and focus on those that are specific relative to undifferentiated hESCs. They find that interacting regions frequently match left ventricle eQTLs and

The data is potentially a very powerful resource, and the integration with GWAS findings is very interesting. However, the analysis and presentation are far too preliminary

Specific points:

Figure 1A. What points are being made here? The authors have chosen to focus on somewhat arbitrary genomic locations, such as “downstream” of genes. Is this enrichment any different from other intergenic locations? What is the interest of analyzing categories such as 5'UTR. Nearly all categories are enriched, except for promoters which have apparently been excluded from the analysis and therefore makes little sense. I would suggest to calculate enrichment of regulatory elements by using more biologically relevant annotations such as public ChromHMM data.

Figure 1b. This figure is very basic and preliminary. Some H3K4me3 enrichment in the interacting regions from active genes is expected, but is this related to promoter-promoter contacts (which have been excluded?), to weak H3K4me3 enrichment in enhancers? Why do the authors only focus on CM-specific genes, and what point do the authors really want to make here?

Figure 2A. The network diagrams are descriptive and not very informative. Instead of integrating the data from all networks to derive general conclusions, the authors provide a diagram of an example of a network, but this does not provide any general insights.

Figure 2B is interesting because it shows some examples for differentiation-specific interactions, but also lacks general insights.

Fig. 3. The photo from a VISTA enhancer is not very interesting on its own, without any context. Perhaps they can show the PCHIC data from this locus. Perhaps they can show a general analysis of the overlap between cPIRS and cardiac VISTA enhancers?

According to the CRISPR experiment this region acts as a repressor. Does this fit the LacZ transgenic data?

Fig. 4 shows interesting findings, but the representation is poor. There is a line for P values, for example, but it does not have a scale.

Fig. 5. This is potentially interesting, but needs much more analysis. The use of ESC interactions as the “expected” distribution is not immediately obvious. Do these results hold true using other expected control distributions? Inflation should be quantified.

Line 138. The conclusions regarding the inflated SNPs with non genome-wide significant P values are not warranted, because they can still represent false positive values. How many SNPs show $P < 10^{-5}$ in that GWAS?

Are established GWAS regions enriched at PIRs?

What do the authors mean when they say that inflated SNPs were “mainly” in cPIRs interacting with promoters of known heart associated genes. This would benefit from some systematic analysis to support this. How much of this was known, what was not known?

What is meant by “the inflated SNPs also coincided with published regions of CCDC141...” Were these regions already known to be genome-wide significant?

Why do authors regard $P < 10^{-5}$ as significant? The paragraph in line 129 starts with “besides the known GWAS regions...”, which implies that the previous text refers to established GWAS regions, which is at odds with that P value threshold.

The authors conclude that “the interactomic map not only enabled us to understand the promoter-enhancer networks...” but they have not established which interactions take place between promoters and enhancers.

Fig 5C would benefit from a schematic which relates SNPs to genes and PIRs

Methods. Most methods involving figure 5 are missing or poorly explained.

Reviewer #4 (Remarks to the Author):

Choy and colleagues presented a new resource of PChi-C data in cardiomyocytes differentiated from human embryonic stem cells. They found 5,126 unique promoter-genome interactions in cardiomyocytes compared to human embryonic stem cells. These cardiomyocyte-specific long-range promoter interactions are enriched by hESC-CM specific promoters, eQTLs, and GWAS, providing valuable information to better understand cardiomyocyte-specific gene regulation mechanisms. I anticipate this study will be a key resource for whom study cardiomyocyte differentiation and related diseases, but I do have some concerns about the current presentation of their data.

Major comments

1. A key value of this study is a generation of PChi-C data in cardiomyocytes. However, the authors did not provide enough details about experimental procedures for PChi-C. They simplified the method as they followed the previous PChi-C protocols, but the authors should include important details including capture probe information, total sequencing read numbers, a fraction of PCR duplicates, capture ratio, etc. Especially, capture efficiencies are vary depending on batches or samples, thus including this information is critical to full utilization of their data in the future.
2. The authors generated three biological replicates for hESC-CM PChi-C data, but the detailed analysis of reproducibility was not provided. They should describe how many interactions are identified from each replicate and how many of them are actually reproducible in all three replicates. Also, they should comment on how their reproducibility rate affects the interpretation of their results.
3. The authors conducted PChi-C analysis using CHICAGO pipeline, which is designed to analyze capture Hi-C data. However, as a user can select many different parameters in CHICAGO pipeline they should provide all parameter information to reproduce their results. Also, it is unclear how they processed PChi-C data using HICUP to make an input of CHICAGO because PChi-C data is essentially different from Hi-C in terms of “capture” step.
4. hESC-CM specific interactions were identified by comparing a previously published PChi-C data on hESC. However, I am not sure whether hESC or hESC-CM PChi-C data is reliable to directly compare to each other. For example, if a capture efficiency of a certain promoter region is different between samples, this makes artifact when they identify sample-specific interactions. They should perform in-depth additional data analysis to justify the identification of hESC-CM specific interactions

5. Figure 1a – the authors concluded as “the 4696 cPIRs were found~” , but to reach such conclusion, they should perform a similar analysis with a control data such as other cell type-specific cPIRs.

6. Figure2a – how do they define directionality in the network visualization using PChi-C result? PChi-C only provides “interaction” information.

7. Figure 3 – According to GTEx result and MGI (<http://www.informatics.jax.org/>), EDNRA/Ednra gene seems to be expressed in cardiovascular-related system. In this aspect, their interpretation of the CRISPR result is somewhat unclear. Do authors want to say the expressed gene can be further up-regulated by deleting enhancer mm172? Or Ednar gene is not expressed in Rat by mm172, but can be expressed by deleting mm172? Is Ednar gene known as not expressed in rat cardiac myoblast cell line?

8. Please provide the list of identified interaction as a table.

Reviewer #1:

Dr Choy & colleagues perform promotor capture Hi-C in human ESC-derived cardiomyocytes, and annotate just over 5k promotor-genome interactions.

Since these interactions are highly cell-type specific, this provides a valuable new annotation of the (human) non-coding genome of relevance and value particularly to the cardiovascular community. The authors demonstrate the value of this cell-specific atlas by annotating relevant GWAS hits and eQTLs, and identifying physical interactions between tagging SNPs and promoters of genes that identify new candidate genes for (some of) the traits analysed.

Comments

My only major concern is that the primary comparison is between newly generated hESC-CM data and historical hESC data, here described by citation only. It looks like these were derived from the same ESC lineage (WA-09), and there is substantial authorship overlap so I presume much of the experimental workflow is directly comparable, but it would be helpful to reproduce a little of the original methodological detail here for the readers convenience and reassurance (e.g. so that it is apparent that these are the same source lineages without referring back to ref 5).

Please also clarify what QC was done to control for any batch effects between libraries prepared at different times (and different place?) or other technical differences that might affect the primary analyses.

We thank the Reviewer for the constructive comments.

We accept the criticism regarding the direct comparison of hESC-CM with previously published hESC data, and now avoid a formal differential calling of interactions between these datasets. Instead, we compare the results for each cell type with its own permuted interaction set, generated according to Javierre et al. Cell, 2016 (please see Methods for details).

Methods - much of the methodology is “as previously described” (refs 3 and 5). I think this could bear some repetition (in condensed form) here for convenience, but leave to your discretion. As above - confirm the origins of the hESC data used for comparison, and methods to control for batch effects.

We included more info for the PCHiC procedure and cell lineages have been made clear in the Methods.

Minor comments

These are mainly comments on data presentation for the authors' consideration

Table S1 - what is the background gene list used for pathway analysis? all genes expressed in hESC-CM?

We included the genes having their transcription start sites in the baits of hESC-CM promoter interactions. Statement “Genes having their transcription start sites in the promoter baits of the top 20% hESC-CM promoter interactions were identified as key regulators of cardiovascular development or functions (Table S2).” has been made in the Text.

Figure 1a — the legend could be more descriptive. In first panel (promotor) are distances either side of promotor (1kb each side), a span centred on the promotor, or upstream only?

Presumably downstream is unidirectional downstream of UTR.

What is the source of the genome annotations used by CEAS? I note that this gives 46.2% of the genome as “genic” (including introns), and 42.4% as intronic, which I think is at the upper limit of the

range of estimates, and will depend on the gene models used for annotations. Suggest give CEAS version and source of annotations.

The CEAS analysis has been excluded from the revised manuscript.

Figure 1b - Personally I do not find the heat map a very helpful representation of 8 numbers here - difficult to compare differences by eye, and plot is not grayscale-safe if printed. Not sure if color palette is red/green color-blind safe?

Thank you for pointing this out. With the new analytical approach, we managed to analyse more histone marks and the heatmap representation makes more sense now. The colour palette is now colour-blind safe.

Figure 2a - top panel: consider adding a track with a color-coded marked that discriminates the promoters from PIRs.

lower panel - could similarly color code PIR vs promotor for clarity. Are the arrows meant to signify directionality (would expect them to be towards the gene promotor if an enhancer/repressor?)?

Figure 2 has been reorganised to Figure 4 of the revised manuscript.

- 1) For the "genome browser view" (Figure 4b), we have tried different colour codes for promoters and PIRs but due to the high amount of interactions in hESC-CM around *MYH6*, the figure was too overlapped and complex to see the colours. The point of the figure was to compare the complexity of MYH6 network in both cell types so single coloured track should be sufficient.
- 2) For the "network view" (Figure 4a), promoter nodes are yellow and labelled but PIR nodes are blue and not labelled. The directionality is arbitrary so we have removed it.

Have you considered providing a URL to a hosted track to make this browsable on UCSC or ensembl? (also fig 2b)

We will make the processed data available in the Capture HiC Plotter <http://www.chicp.org> after the paper is published.

line 81 - "a number of cardiac VISTA enhancers were found to overlap" - what proportion of your PIR overlap with VISTA enhancers, and are these statistically enriched? A brief description of the method used to define VISTA enhancers, to contrast with the methods used here, might be useful for some readers.

We have included Figure 2a to show that the overlap was significant. We took the VISTA enhancers from its browser so we follow their definition in their publications.

I would add a comment acknowledging that the only region selected for validation in the wet lab was also annotated by VISTA - a broader set of wet lab validations would give better sense of whether there is an important false positive rate (though enrichment analyses provide evidence for biological relevance)

Part one of the comment – done. It has been made clear that mm172 is a VISTA enhancer.

Part two – We accept the point and have indeed started to do a broader set of wet lab validations. However we would contend that this is a subject for a different publication.

line 92 - "edna" should be "ednra"

The typo has been corrected.

line 101 - though whole blood eQTLs are only "marginally significant", they are still enriched - and presumably this was anticipated as a negative control. Suggest expanding the comment here (e.g. after bonferroni correction it more enriched than atrial tissue that would be expected to be more similar).

With the new analytical approach, this issue is resolved.

Data & code availability - The accession GSE100720 is private and scheduled for release on 3rd July 2020. I did not receive a reviewer's secure token to preview this material, so have not reviewed this content.

Both the raw and processed data should be available for reproducibility: I am not sure whether both have been deposited.

The code should also be available along with the data - I am not sure whether GEO supports this - if not then would suggest using a GitHub "release" or equivalent. This can be released "as is" in its working form without any special presentation for publication.

We apologise for this omission. The token is "czgvkywobpcxpsd".

Reviewer #2:

This is a concise report presenting new promoter capture Hi-C data (PChi-C) in cardiomyocytes derived from human embryonic stem cells (hESC-CM). The authors show that many of the promoter-interacting DNA segments (PIRs) specific to hESC-CM (cPIRs) act as distal regulatory elements of the target genes inferred from the promoter interactions. Furthermore, the authors used the PIRs to focus on SNPs associated with cardiac phenotypes via GWAS, providing a better prediction of likely target genes and enabling discovery of significant association of SNPs at a locus not previously associated with cardiac conductance and rhythm disorders. This use of promoter interaction data to filter potentially interesting SNPs reduces the multiple hypothesis testing burden and allows discovery of associations that fall below genome-wide significance. The authors make a strong point that the approach followed in this paper should be broadly applicable.

The following are points for improving the manuscript.

We thank Professor Hardison for his constructive comments. As mentioned above (response to Reviewer #1's comments), we now avoid a direct calling of differential interactions between the two cell types, and instead compare each dataset with its own permuted control.

(1) p. 1, lines 18-19: Characterized enhancers are genomic regulators, but eQTLs and tag SNPs associated with traits by GWAS are genetic variants potentially associated with regulatory elements.

We have rewritten the abstract to avoid the problem.

(2) The authors should point out that candidates for distal gene regulatory elements (ascertained by epigenetic features such as chromatin accessibility and histone modifications) are enriched for phenotype-associated SNPs discovered via GWAS, with appropriate references.

Statement "Phenotype-associated SNPs discovered through genome-wide association study (GWAS) are enriched in regulatory elements ascertained by epigenetic features such as chromatin accessibility and histone modifications." has been made to initiate the discussion on the GWAS analysis with reference 22.

(3) Fig. 1b: These results would be displayed more clearly as a graph of enrichment and depletion rather than a heat map. The legend says that the heat maps shows the enrichment of hESC-CM specific H3K4me3 enrichment in promoter, but it is not clear what is being plotted for the cPIRs.

With the new analytical approach, we managed to analyse more histone marks and therefore heat maps seem to be an appropriate representation for the amount of info to be presented. We mentioned "and their corresponding cPIRs" for the promoters.

(4) Fig. 2a and 2b: In the genome browser views, the positions of promoter vs. cPIR nodes are not clear. Showing the "promoter" DNA fragment "baits" would help, but it would be best to also include distinctive icons for promoter vs cPIR.

Figure 2 has been reorganised to Figure 4 of the revised manuscript.

1) For the "genome browser view" (Figure 4b), we have tried different colour codes for promoters and PIRs but due to the high amount of interactions in hESC-CM around MYH6, the figure was too overlapped and complex to see the colours. The point of the figure was to compare the complexity of MYH6 network in both cell types so single coloured track should be sufficient.

2) For the “network view” (Figure 4a), promoter nodes are yellow and labelled but PIR nodes are blue and not labelled. The directionality is arbitrary so we have removed it.

(5) p. 3, line 90: Presumably the authors deleted the rat ortholog of mm172.

“Rat ortholog” has been included.

(6) p. 5, lines 159-161: The authors state that the hESC-CM specific promoter interactome has a higher success rate in mapping target genes of particular eQTLs, but they should state the method or resource to which they are comparing.

Compared to random backgrounds with the new analytical approach. The method has been included in Methods

Signed,
Ross Hardison, Penn State University

Reviewer #3:

The authors performed promoter capture Hi-C (PCHiC) in cardiomyocytes differentiated from human embryonic stem cells (hESC-CMs) to identify distal promoter contacts, and focus on those that are specific relative to undifferentiated hESCs. They find that interacting regions frequently match left ventricle eQTLs and

The data is potentially a very powerful resource, and the integration with GWAS findings is very interesting. However, the analysis and presentation are far too preliminary

Specific points:

Fig. 5. This is potentially interesting, but needs much more analysis. The use of ESC interactions as the “expected” distribution is not immediately obvious. Do these results hold true using other expected control distributions? Inflation should be quantified.

We thank the Reviewer for the constructive comments. As mentioned above (response to Reviewer #1’s comments), we now avoid a direct calling of differential interactions between the two cell types, and instead compare each dataset with its own permuted control. The QQ plots are provided with inflation factors.

Figure 1A. What points are being made here? The authors have chosen to focus on somewhat arbitrary genomic locations, such as “downstream” of genes. Is this enrichment any different from other intergenic locations? What is the interest of analyzing categories such as 5’UTR. Nearly all categories are enriched, except for promoters which have apparently been excluded from the analysis and therefore makes little sense. I would suggest to calculate enrichment of regulatory elements by using more biologically relevant annotations such as public ChromHMM data.

The CEAS analysis has been excluded from the revised manuscript.

Figure 1b. This figure is very basic and preliminary. Some H3K4me3 enrichment in the interacting regions from active genes is expected, but is this related to promoter-promoter contacts (which have been excluded?), to weak H3K4me3 enrichment in enhancers? Why do the authors only focus on CM-specific genes, and what point do the authors really want to make here?

With the new analytical approach, we managed to analyse more histone marks and found stronger associations. We focus on all the genes sub-classified to expression quartiles. Promoter-promoter interactions are not included in this study and we are trying to show that histone marks of a cPIR reflect the transcriptional status of its interacting promoter just like what Javierre et al. Cell, 2016 reported.

Figure 2A. The network diagrams are descriptive and not very informative. Instead of integrating the data from all networks to derive general conclusions, the authors provide a diagram of an example of a network, but this does not provide any general insights.

Figure 2B is interesting because it shows some examples for differentiation-specific interactions, but also lacks general insights.

Figure 2 has been reorganised to Figure 4 of the revised manuscript. The Figure shows that *MYH6*, one of the important cardiac genes, forms the most complex network in the top 20% hESC-CM’s promoter interactions. This network was more complex than the equivalent network in the hESC. This is a very significant finding for cardiologists to understand the biological relevance of promoter interactions in cardiomyocytes.

Fig. 3. The photo from a VISTA enhancer is not very interesting on its own, without any context. Perhaps they can show the PCHIC data from this locus. Perhaps they can show a general analysis of the overlap between cPIRS and cardiac VISTA enhancers?

We have included promoter-cPIR interactions of EDNRA (with the VISTA mm172 interaction highlighted) in Figure 2.

According to the CRISPR experiment this region acts as a repressor. Does this fit the LacZ transgenic data?

We have included the explanation in the Discussion: "The repressive nature of this cPIR seems to fit in the paradigm of repressed lineage-specifying promoter interactions at the pluripotent state (ref 2). However, silencers of the same region could be disrupted as well when a cPIR/enhancer is targeted by CRISPR-Cas9 since enhancers and silencers can be located in the same locus or clustered in the same locus control region (LCR) (ref 40)."

Fig. 4 shows interesting findings, but the representation is poor. There is a line for P values, for example, but it does not have a scale.

We use dots for P values referring to right y-axis ($-\log_{10}(\text{p-value})$) now (Figure 2a and 3). The 0.05 threshold line is referring to right y-axis ($-\log_{10}(\text{p-value})$) as well.

Fig. 5. This is potentially interesting, but needs much more analysis. The use of ESC interactions as the "expected" distribution is not immediately obvious. Do these results hold true using other expected control distributions? Inflation should be quantified.

This critical point has prompted a full reanalysis of the data, thank you. We now avoid a direct calling of differential interactions between the two cell types, and instead compare each dataset with its own permuted control. The QQ plots are provided with inflation factors.

Line 138. The conclusions regarding the inflated SNPs with non genome-wide significant P values are not warranted, because they can still represent false positive values. How many SNPs show $P < 10^{-5}$ in that GWAS? Are established GWAS regions enriched at PIRs?

We used GoShifter (Trynka et al. Am J Hum Genet, 2015) to check if inflated SNPs overlap cPIRS significantly. Inflated SNPs of heart-rate GWAS were found to be significantly enriched in cPIRS.

What do the authors mean when they say that inflated SNP were "mainly" in cPIRs interacting with promoters of known heart associated genes. This would benefit from some systematic analysis to support this. How much of this was known, what was not known?

The statement has been re-written as: "Some of the potentially significant SNPs (at suggestive threshold of $p < 1 \times 10^{-5}$) associated with CRD and located in cPIRs were interacting with promoters of known heart-rate associated genes (ref 26) such as CCDC141 on chromosome 2, GJA1 on chromosome 6 and MYH6 on chromosome 14."

What is meant by "the inflated SNPs also coincided with published regions of CCDC141..." Were these regions already known to be genome-wide significant?

Yes in the original GWAS paper den Hoed et al. Nat Genet, 2013. Reference has been included in the text.

Why do authors regard $P < 10^{-5}$ as significant? The paragraph in line 129 starts with “besides the known GWAS regions...”, which implies that the previous text refers to established GWAS regions, which is at odds with that P value threshold.

We do not regard $p < 10^{-5}$ as significant, we believe it is appropriate to refer to it as “suggestive” – indeed regions with this degree of support in discovery samples have achieved replication in previous studies. To avoid confusion we have removed the phrase “besides the known GWAS regions...”. We aim to draw attention to the fact that interactomic data could potentially be used to assist in elucidating “missing heritability” due to biologically significant effects too small to be detected in GWAS samples to date (or feasibly assembled, for some conditions).

The authors conclude that “the interactomic map not only enabled us to understand the promoter-enhancer networks...” but they have not established which interactions take place between promoters and enhancers.

“Promoter-enhancer” has been replaced by “promoter-genome”.

Fig 5C would benefit from a schematic which relates SNPs to genes and PIRs

Methods. Most methods involving figure 5 are missing or poorly explained.

We have re-written the Methods to include all necessary info for Figure 5 analyses.

Reviewer #4 (Remarks to the Author):

Choy and colleagues presented a new resource of PCHI-C data in cardiomyocytes differentiated from human embryonic stem cells. They found 5,126 unique promoter-genome interactions in cardiomyocytes compared to human embryonic stem cells. These cardiomyocyte-specific long-range promoter interactions are enriched by hESC-CM specific promoters, eQTLs, and GWAS, providing valuable information to better understand cardiomyocyte-specific gene regulation mechanisms. I anticipate this study will be a key resource for whom study cardiomyocyte differentiation and related diseases, but I do have some concerns about the current presentation of their data.

We thank the Reviewer for the constructive comments.

Major comments

1. A key value of this study is a generation of PCHI-C data in cardiomyocytes. However, the authors did not provide enough details about experimental procedures for PCHI-C. They simplified the method as they followed the previous PCHI-C protocols, but the authors should include important details including capture probe information, total sequencing read numbers, a fraction of PCR duplicates, capture ratio, etc. Especially, capture efficiencies are vary depending on batches or samples, thus including this information is critical to full utilization of their data in the future.

We have included additional methological info in the text. We now avoid a direct calling of differential interactions between the two cell types, and instead compare each dataset with its own permuted control.

2. The authors generated three biological replicates for hESC-CM PCHI-C data, but the detailed analysis of reproducibility was not provided. They should describe how many interactions are identified from each replicate and how many of them are actually reproducible in all three replicates. Also, they should comment on how their reproducibility rate affects the interpretation of their results.

This analysis has limited utility due to the fact that PCHI-C data are significantly undersampled (sparse), leading to reduced sensitivity, which drives down the observed overlap between individual replicates. In other words, and somewhat counterintuitively, the incomplete overlap between PCHI-C data replicates is primarily driven by false-negative observations in each replicate, as opposed to false-positive ones. We respectfully refer the Reviewer to the paper describing the CHiCAGO algorithm (Cairns et al., Genome Biology 2016), and particularly to analyses presented in Figure S4 in that study, for more detail. Therefore, the CHiCAGO pipeline combines signals across multiple replicates (after normalisation for effective library size) to maximise sensitivity and power.

3. The authors conducted PCHI-C analysis using CHICAGO pipeline, which is designed to analyze capture Hi-C data. However, as a user can select many different parameters in CHICAGO pipeline they should provide all parameter information to reproduce their results. Also, it is unclear how they processed PCHI-C data using HiCUP to make an input of CHICAGO because PCHI-C data is essentially different from Hi-C in terms of “capture” step.

HiCUP can be used with both Hi-C and Capture Hi-C data, as explicitly stated in the paper describing this tool (Wingett et al., F1000Res 2015). CHiCAGO was run with default parameters (Methods updated).

4. hESC-CM specific interactions were identified by comparing a previously published PCHI-C data on hESC. However, I am not sure whether hESC or hESC-CM PCHI-C data is reliable to directly compare to each other. For example, if a capture efficiency of a certain promoter region is different between

samples, this makes artifact when they identify sample-specific interactions. They should perform in-depth additional data analysis to justify the identification of hESC-CM specific interactions

5. Figure 1a – the authors concluded as “the 4696 cPIRs were found~” , but to reach such conclusion, they should perform a similar analysis with a control data such as other cell type-specific cPIRs.

Other reviewers also made this critical point, which has prompted a complete reanalysis of the data. Thank you. As mentioned above (response to Reviewer #1’s comments), we now avoid a direct calling of differential interactions between the two cell types, and instead compare each dataset with its own permuted control. The QQ plots are provided with inflation factors.

6. Figure2a – how do they define directionality in the network visualization using PCHI-C result? PCHI-C only provides “interaction” information.

The directionality is arbitrary so we have removed it.

7. Figure 3 – According to GTEx result and MGI (<http://www.informatics.jax.org/>), EDNRA/Ednra gene seems to be expressed in cardiovascular-related system. In this aspect, their interpretation of the CRISPR result is somewhat unclear. Do authors want to say the expressed gene can be further up-regulated by deleting enhancer mm172? Or Ednar gene is not expressed in Rat by mm172, but can be expressed by deleting mm172? Is Ednar gene known as not expressed in rat cardiac myoblast cell line?

We have included the explanation in the Discussion: “The repressive nature of this cPIR seems to fit in the paradigm of repressed lineage-specifying promoter interactions at the pluripotent state (ref 2). However, silencers of the same region could be disrupted as well when a cPIR/enhancer is targeted by CRISPR-Cas9 since enhancers and silencers can be located in the same locus or clustered in the same locus control region (LCR) (ref 40).” EDNRA was expressed in rat cardiac myoblasts but not highly.

8. Please provide the list of identified interaction as a table.

We have included the list as Table S1.

Reviewer #2 (Remarks to the Author):

In the revised version, the authors made many changes to the text and figures that clarify and expand the results. The revised manuscript addresses all my concerns and more.

Reviewer #3 had no further comments to the authors.

Reviewer #4 (Remarks to the Author):

The revised manuscript has been greatly improved. However, taking into account the impacts and values of resources in this study, the authors should address the following issues before publication.

1. The description of data processing of pcHi-C data is too short to understand the details. What is the percentage of captured reads and off-target reads? Did the authors remove off-target reads during downstream analysis? How should the authors consider various capture efficiencies in the promoter regions when they call interaction peaks.
2. It is difficult to agree that pcHi-C data is significantly undersampled. In fact, the main purpose of pcHi-C is oversampling of promoter-centered chromatin interactions compared to Hi-C method. Although the authors explained that the incomplete overlap between biological replicates is primarily induced by false-negative observation, it is still important to show reproducibility of biological replicates as they are.

Reviewer #5 (Remarks to the Author):

Comments to the Authors:

The main concern raised regarding the direct comparison of the hESC and hESC-CM data, that was confounded by batch, has now been appropriately addressed by removing the analysis on the combined dataset and instead comparing each cell type to a permuted set of interactions and

comparing those results across cell types to characterize hESC and hESC-CM similarities and differences.

We have addressed the requests of Reviewer #4

1. *The description of data processing of pcHi-C data is too short to understand the details. What is the percentage of captured reads and off-target reads? Did the authors remove off-target reads during downstream analysis? How should the authors consider various capture efficiencies in the promoter regions when they call interaction peaks.*

We have included the information regarding capture efficiencies and mentioned that only on-target reads were used in the Results section. Capture efficiencies are reflected, among other parameters, in the so-called bait scaling factors (s_j) that Chicago calculates in a data driven way as described in Cairns *et al.*

2. *It is difficult to agree that pcHi-C data is significantly undersampled. In fact, the main purpose of pcHi-C is oversampling of promoter-centered chromatin interactions compared to Hi-C method. Although the authors explained that the incomplete overlap between biological replicates is primarily induced by false-negative observation, it is still important to show reproducibility of biological replicates as they are.*

We have included a 3-way Venn diagram (Supplementary Fig. 1) to show the overlaps of promoter interactions in the three biological replicates.

3. We have also reformatted the manuscript, figures and supplementary information according to editorial suggestions. Especially uncropped images of PCR gels and immunoblots have been included as Supplementary Fig. 2 and 3.